

**Field Observations Reveal Substantially Higher Scattering Refractive**
**Index in Secondary Versus Primary Organic Aerosols**
Junlin Shen[1], Li Liu[2], Fengling Yuan[1], Biao Luo[1], Hongqing Qiao[1], Miaomiao Zhai[1], Gang Zhao[3],
Hanbing Xu[4], Fei Li[5], Yu Zou[2], Tao Deng[2], Xuejiao Deng[2], Ye Kuang[1]
[1] Institute for Environmental and Climate Research, College of Environment and Climate, Jinan
University, Guangzhou, 511443, Guangdong, China
[2] Guangzhou Institute of Tropical and Marine Meteorology of China Meteorological Administration,
GBA Academy of Meteorological Research, Guangzhou, 510640,  China.
[3] Key Laboratory of Ecology and Environment in Minority Areas, Minzu University of China, National
Ethnic Affairs Commission, Beijing,100081, China
[4] School of Computer Science and Engineering, Sun Yat-Sen University, Guangzhou, 510275,
Guangdong, China
[5] Xiamen Key Laboratory of Straits Meteorology, Xiamen Meteorological Bureau, Xiamen, 361012,
Fujian, China
Correspondence: Ye Kuang (kuangye@jnu.edu.cn) and Li Liu (liul@gd121.cn)





**Abstract:**

Aerosol-radiation interactions play a crucial role in air pollution and climate change with scattering being the dominant process. The complex refractive index of organic aerosols is essential for accurately simulating these interactions, with the scattering capability is predominantly determined by the real part of the refractive index ($m_r$). Prevailing models often assume a constant $m_r$ for organic aerosols (e.g., 1.53 or 1.45) at different wavelengths or claim that $m_r$ of primary organic aerosols (POA) is substantially higher than that of secondary organic aerosols (SOA) (e.g., 1.63 for POA and 1.43 for SOA), largely due to a lack of direct measurements. This study employs direct measurements from the DMA-SP2 system to demonstrate a strong diameter dependence of dry state $m_r$ at 1064 nm, closely associated with primary aerosol emissions and secondary aerosol formation. Source apportionment of aerosol size distributions reveals that the $m_r$ of SOA is substantially higher than that of POA. Optical closure calculations, based on extensive dry state observations of aerosol scattering at 525 nm, size distributions, and chemical compositions, confirm this finding. These results challenge existing model assumptions. In addition, further analysis reveals $m_r$ of SOA increases with oxidation level, which is contrary to results of most laboratory studies on evolution of $m_r$ of SOA, which is likely associated with multiphase SOA formation. Our analysis recommends $m_r$ values at 525 nm of 1.37 for POA and 1.59 for SOA. These findings underscore that current modeling practices may introduce substantial inaccuracies in estimating the radiative effects of organic aerosols.



## 1. Introduction

Aerosol-radiation interactions play a crucial role in air pollution and climate change. Atmospheric aerosols scatter and absorb solar radiation, which can alter the thermal structure of the atmosphere, the radiative energy balance of the Earth-atmosphere system, and affect atmospheric visibility. On one hand, aerosols influence the thermal structure of the surface and atmosphere, which affects the development of the atmospheric boundary layer (Zhong et al., 2019), thereby playing an important role in the evolution of pollution processes and air pollution. On the other hand, by scattering and absorbing solar radiation, aerosols can affect the radiative energy balance of the Earth-atmosphere system, impacting both local and global climates. Due to the complexity of atmospheric aerosol components, the direct radiative effect of aerosols (referred to as aerosol-radiation interactions in the latest IPCC report) is the second-largest source of error in accurately assessing anthropogenic climate forcings (IPCC,AR6, 2023) and is a significant factor limiting the accurate prediction of global climate change.

The inability to accurately characterize the complex refractive index of organic aerosols is one of the major sources of error in accurately simulating the direct radiative effects of aerosols (Redemann et al., 2000;Li et al., 2021;Tsigaridis and Kanakidou, 2018). Organic aerosols are a significant component of atmospheric aerosols, on average accounting for about 20-60% of submicron aerosols in most continental regions (Zhang et al., 2007), and in some areas, such as tropical rainforest regions, this proportion can be as high as 90% (Kanakidou et al., 2005). Therefore, organic aerosols are one of the main contributors to the direct radiative effects of aerosols and likely a major source of error in accurately assessing these effects (Moise et al., 2015). Compared to inorganic aerosols, the complex chemical composition of organic aerosols poses a core challenge to accurately quantifying their optical properties(Wu et al., 2021). Based on Mie scattering theory, the core parameters affecting aerosol optical properties are aerosol size and complex refractive index. In both climate models and atmospheric chemical transport models, the complex refractive index is a fundamental parameter for calculating key optical parameters such as the extinction coefficient, single scattering albedo, and asymmetry factor (Moise et al., 2015). The real part ($m_r$) of the complex refractive index corresponds mainly to scattering properties, while the imaginary part corresponds mainly to absorption properties. The extinction of solar radiation by aerosols is determined by aerosol scattering and absorption with scattering being the dominant process (Moise et al., 2015), and accurately characterizing $m_r$ of organic aerosols ($m_{r,OA}$) is thus key to accurately simulating aerosol radiative effects (Li et al., 2021;McMeeking et al., 2005). The review by Tsigaridis and Kanakidou (2018) pointed out that




existing models either treat $m_{r,OA}$ as a constant or treat $m_r$ for POA and SOA ($m_{r,POA}$ and $m_{r,SOA}$) as
constants. For example, Curci, et al. (2019) set $m_{r,POA}$ to 1.63 and $m_{r,SOA}$ to 1.43 in their model.

However, results from existing literature of laboratory studies demonstrate that $m_{r,OA}$ varies a lot.

The advantage of laboratory studies is that they can produce aerosol systems containing only organic
components, with a relatively narrow size range, allowing the retrieval of $m_{r,OA}$ based on scattering
or extinction measurements. Consequently, laboratory quantitative studies on $m_{r,SOA}$ have been
conducted broadly (Moise et al., 2015), while $m_{r,POA}$ are rarely investigated. The results show that
the $m_{r,SOA}$ varies mainly in the range of 1.36-1.66, and the variation of $m_{r,SOA}$ is closely related to its
precursors and oxidation pathways (Moise et al., 2015;Kim et al., 2014;Lambe et al., 2013;He et al.,
2018). For example, results of He et al. (2018) demonstrate that $m_{r,SOA}$ first increase with the oxidation
state parameter O/C and then decrease with O/C during the aging. Li et al. (2023b) further developed
a parameterization scheme for $m_{r,SOA}$ based on O/C and H/C and validated it using laboratory
experiment results, however, its applicability to POA and SOA on the basis of field measurements
remains lacking. Overall, the difficulty in direct quantification of $m_{r,OA}$ on the basis of field
measurements has made the variation characteristics of $m_{r,OA}$ in the atmosphere remain elusive.

In this study, using field measurements of aerosol refractive index, aerosol size distributions,

aerosol scattering properties as well as aerosol chemical compositions, the remarkable difference in
$m_{r,POA}$ and $m_{r,SOA}$ is revealed, which serves strong observational evidence that $m_{r,SOA}$ is much higher
than $m_{r,POA}$ and values for model settings of $m_{r,POA}$ and $m_{r,SOA}$ are recommended on the basis of
observations.
**2. Materials and Methods**
**2.1 Field measurements**

In this study, we utilized datasets from two field campaigns conducted at Haizhu Wetland Park,

Guangzhou, China. The first campaign lasted less than two months, from January 12 to February 27,
2022, while the second was a longer-term campaign spanning approximately seven months, from July
27, 2022, to February 28, 2023.

During the first campaign, we observed the particle number size distribution (PNSD) in a dry

state (relative humidity for indoor measurements was near 10%), ranging from 13 nm to 800 nm, using
a Scanning Mobility Particle Sizer (SMPS, model 3086 and particle counter 3776 from TSI) with a
temporal resolution of 5 minutes. The $m_r$ of BC-free aerosols with diameters of 235 nm, 270 nm, 300
nm, 335 nm, 370 nm, and 400 nm were measured using the DMA-SP2 system (differential mobility



analyzer in tandem with single-particle soot photometer). This measurement method for $m_r$ was
previously proposed by Zhao et al. (2019c). To briefly explain the $m_r$ measurement using the DMA-
SP2 system: the SP2 channels receive both scattering and incandescent signals from sampled aerosols.
For pure scattering aerosols, the peak of the scattering signal is positively correlated with aerosol
scattering ability, which is determined by aerosol size and $m_r$ for spherical particles. The scattering
strength (S) at 1064 nm can be expressed as:
$S = C \times I_0 \times \sigma \times (PF_{45°} + PF_{135°}),$
Where $I_0$ is the instrument's laser intensity, C is a constant determined by the instrument's response
characteristics, σ is the scattering coefficient of aerosols, $PF_{45°}$ and $PF_{135°}$ are the scattering phase
functions at 45° and 135°, respectively. The relationship between the peak of the scattering signal and
the scattering strength of pure scattering aerosols has been calibrated using ammonium sulfate (see
Sect. S1 of the supplement). Consequently, the $m_r$ at 1064 nm ($m_{r1064}$) of SP2 laser for pure scattering
aerosols can be retrieved using the particle size from the DMA and the scattering strength from the
SP2, following the method demonstrated by Zhao et al. (2019c). The DMA-SP2 technique offers the
advantage of providing direct measurements of $m_{r1064}$. However, it also has certain limitations. For
instance, in this study, the $m_{r1064}$ measurements are constrained to a diameter range of approximately
235 to 400 nm, depending on laser intensity, thereby excluding smaller particles (<200 nm) and
relatively larger submicron particles (>400 nm). Additionally, some BC-free particles exhibit
absorptive properties, such as brown carbon containing particles that may absorb at infrared
wavelengths (Hoffer et al., 2017). Thus, these particles may absorb laser energy during scattering
measurements, causing heating that can lead to the evaporation of semi-volatile or even low-volatile
species from the particle phase, potentially biasing the $m_{r1064}$ measurements, although this effect is
likely very small because this type of brown carbon aerosols likely account for very small portions of
BC-free aerosols (Luo et al., 2022). Non-refractory submicron (NR-PM$_1$) aerosol chemical
compositions—including ammonium (NH4), nitrate (NO3), sulfate (SO4), chloride (Cl), and organic
components—were measured using a Quadrupole Aerosol Chemical Speciation Monitor (Q-ACSM).
Details about the quality assurance of Q-ACSM measurements during this campaign are provided by
Li et al. (2023a).

During the second campaign, direct measurements of $m_{r1064}$ were not conducted. Instead, dry

state aerosol scattering coefficients of total suspended particles (TSP) at 450, 525, and 635 nm were
measured using a nephelometer under nearly dry conditions (below 15% relative humidity). The dry-
state (relative humidity below 20%) PNSD, ranging from 13 to 800 nm, was again measured using the



SMPS. Additionally, multi-wavelength aerosol absorption measurements were performed using an
Aethalometer (AE$_{33}$, (Drinovec et al., 2015)), and the NR-PM$_1$ aerosol chemical compositions were
also measured using the Q-ACSM.

**2.2 Source Analysis Methods of Organic Aerosols and Aerosol Size Distributions**

The multilinear engine (ME-2) technique (Canonaco et al., 2013;Canonaco et al., 2021) was
applied to the organic aerosol mass spectra to resolve the sources of organic aerosols. Multilinear
Engine (ME-2) is an upgrade of widely used Positive Matrix Factorization (PMF) technique and runs
on an IGOR-based interface (Canonaco et al., 2013). Different from traditional PMF, ME-2 offers
capability of constraining the spectra variation extent of OA factor with given priori mass spectra
(Canonaco et al., 2013;Guo et al., 2020). Four factors were identified across both field campaigns: two
primary organic aerosol (POA) factors and two secondary organic aerosol (SOA) factors. The POA
factors consisted of hydrocarbon-like organic aerosol (HOA) and cooking-like organic aerosol (COA),
while the SOA factors consisted of less oxygenated organic aerosol (LOOA) and more oxygenated
organic aerosol (MOOA). The spectral profiles of HOA and COA obtained in Liu et al. (2022) were
used in the ME-2 procedure to constrain POA factor variations.
The POA factors exhibited consistent spectral profiles and elemental ratios between the two
campaigns. For example, the O/C ratios of HOA were 0.16 and 0.17, and the O/C ratios of COA were
0.12 and 0.14, respectively. However, the resolved SOA factors differed between the campaigns. The
O/C ratio of LOOA in the short-term campaign was 0.89, while it was 0.60 in the long-term campaign.
Similarly, the O/C ratio of MOOA was 0.93 in the short-term campaign and 1.27 in the long-term
campaign. These differences do not affect the overall analysis of this study, as the focus is primarily
on the distinction between POA and SOA. More details about the source analysis of organic aerosols
can be found in the supplements of Li et al. (2023a) for the short-term campaign and Qiao et al. (2024)
for the long-term campaign.
Additionally, following the positive matrix factorization (PMF) procedure for the PNSD
measurements and the source apportionment method introduced by Cai et al. (2020a), five PNSD
factors were identified (PMF 2, ver. 4.2, 111 bins for PNSD ranging from 14 nm to 736 nm as inputs).
For details on the determination of the factor numbers and the PNSD factor analysis, please refer to
Sect. S2 of the supplement. In the source apportionment of PNSD factors, ammonium, nitrate, and
sulfate measurements were paired as ammonium sulfate (AS) and ammonium nitrate (AN) using the
scheme proposed by Gysel et al. (20) considering that different impacts of AS and AN formation on
PNSD. The mass concentrations of refractory black carbon (rBC) during the short-term campaign were





integrated from size-resolved rBC measurements obtained using the DMA-SP2 system, as described
in Li et al. (2023a). Correlation analysis between mass concentrations of OA factors, rBC, AS as well
as AN and resolved PNSD factors were performed to help explore sources of different PNSD factors.
In addition, the densities of aerosol species used for volume calculations in this study were
consistent with those in Kuang et al. (2021): 1.78 g/cm³ for AS and AN, 1.0 g/cm³ for HOA and COA,
1.2 g/cm³ for LOOA, and 1.4 g/cm³ for MOOA. However, 1.0 g/cm³ was chosen for rBC on the basis
of previous observations results (Zhang et al., 2016b;Zhao et al., 2020;Zhou et al., 2022). Calculating
BC volume for hygroscopicity requires the material density of BC, as described in Kuang et al. (2021).
However, for Mie calculations in this study, effective BC density is needed to determine the BC core
size. Since the presence of air voids (Zhang et al., 2016b;Zhao et al., 2020) within BC particles
increases their apparent size compared to calculations based on the material density. The source
analysis of resolved PNSD factors through combination of ACSM measurements was discussed
comprehensively in Sect 3.1 to help explore observed $m_r$ diameter dependence.

**2.3 Optical Closure and associated $m_r$ Retrieval**

During the second long-term field campaign, dry-state aerosol scattering coefficients of TSP and
PNSD (ranging from 13 to 800 nm, generally covering dry-state PM1), as well as black carbon (BC)
mass concentrations, were simultaneously measured, making it feasible to perform a closure between
the measured and simulated aerosol scattering ($\sigma_{sp,obs}$ vs $\sigma_{sp,sim}$). Details about aerosol scattering
calculating procedures and simulations using the Mie code of BHCOAT (Bohren and Huffman,
1998;Cheng et al., 2009) could be found in Sect. S4. Five key issues needed to be addressed for this
closure: (1) The size range mismatch between aerosol scattering measurements (TSP) and PNSD
measurements (dry-state PM1); (2) BC mass size distributions and mixing state, and other BC related
parameters such as density and refractive index; (3) The $m_r$ of BC-free aerosols at 525 nm; (4)
Imaginary part of BC-free aerosols which is mostly associated with brown carbon; (5) Corrections for
integrating nephelometer measurements to account for truncation errors and light source non-idealities
(Müller et al., 2011).
Recent field observations (detailed in Sect. S3 of the supplement), utilizing a system (Kuang et
al., 2024) that coupled different aerosol inlets with a integrating nephelometer, demonstrated that in
Guangzhou's urban area, scattering coefficients of dry-state PM$_1$ and TSP ($\sigma_{sp,PM_1}$ vs $\sigma_{sp,TSP}$)
generally agree well (Fig. S4a, R = 0.99). However, their ratio varies substantially depending on
aerosol scattering levels (Fig. S4b). Specifically, the ratio $\sigma_{sp,TSP}/\sigma_{sp,PM_1}$ at 525 nm exceeds 1.2 when
$\sigma_{sp,TSP}$ is below 50 Mm⁻¹, reaching approximately 1.5 when $\sigma_{sp,TSP}$ is around 10 Mm⁻¹. This ratio



decreases as $\sigma_{sp,TSP}$ increases and stabilizes (near 1.08) when $\sigma_{sp,TSP}$ exceeds 90 Mm⁻¹. Consequently,
for the closure, measured dry-state $\sigma_{sp,TSP}$ was corrected to dry-state $\sigma_{sp,PM_1}$ using the observed
$\sigma_{sp,TSP}$-dependent relationship shown in Fig. S4b (ratio of 1.08). This ratio may vary across seasons.
However, aerosol scattering simulation results based on particle size distribution measurements that
cover the supermicron range, conducted during six campaigns across various locations and seasons in
the North China Plain, indicate that PM₁ on average contributes approximately 90% to TSP scattering
(Fig. 2 of Kuang et al. (2018)). This aligns closely with the average ratio of 1.08 determined in this
study through direct scattering measurements, suggesting that this ratio likely does not vary
substantially.
The BC mass size distributions and mixing state during the first short-term campaign were
analyzed systematically in a previous study conducted by Li et al. (2023a). Two key findings emerged:
(1) BC mass size distributions for diameters >100 nm could be represented by a single lognormal mode,
with a geometric mean diameter ($D_g$) of 258 (±16) nm and a geometric standard deviation ($\sigma_g$) of 1.69;
(2) Nearly half of the BC mass was identified as externally mixed. The mass fraction of externally
mixed BC in total BC ($R_{ext}$) was calculated as $0.56 \pm 0.16$, and the number fraction ($R_{csm}$) of internally
mixed BC (represented by the core-shell model) in total number of internally mixed BC and BC-free
particles was $0.13 \pm 0.12$. These findings indicated that, despite clear evidence of secondary aerosol
formation during the first campaign, BC mass size distributions and mixing states varied within a
relatively narrow range, primarily influenced by traffic emissions (Li et al., 2023a). Therefore, the
parameters $D_g$= 258 nm, $\sigma_g$= 1.69, $R_{ext}$= 0.56, and $R_{csm}$= 0.13 were used to distribute the BC mass
concentrations measured by the AE₃₃ and account for BC mixing states. In brief, aerosol particles were
divided into three types: externally BC, internally mixed BC with BC as the core, and BC-free. BC
mass was distributed into different diameters using the established lognormal function, and further
allocated to externally and internally mixed BC-containing particles using parameters $R_{ext}$ and $R_{csm}$.
Note that these values regarding BC size distributions and mixing states are not expected to remain
constant throughout the campaign. However, sensitivity test conducted on the basis of observations in
the short campaign help boost the confidence. Sensitivity test results shown in Fig.S5 show that even
if the geometric mean diameter of the BC mass size distribution changes from 180 nm to 600 nm, very
large variation according to reported distribution in literatures (Zhao et al., 2019b), the relative changes
in scattering calculations remain relatively small (~2%). Instead, the BC mixing state plays a more
critical role. For example, changing the mixing state from completely externally mixed to a fully core-
shell internal mixture results in changes of approximately 10%. However, such a scenario represents
an extreme condition. Considering the observation site is located near BC source regions, BC aerosols





are likely closer to being externally mixed. Errors associated with the BC mixing state parameter are
estimated to be less than 2.5%. This inference assumes that BC mixing states in this region vary from
completely externally mixed to half externally mixed (much larger than observed in the first short
campaign). Using an average value in this case would result in uncertainties of less than 2.5%. Errors
associated with BC mixing state would likely be smaller. In addition, other BC related parameters
might also induce retrieval errors, such as uncertainties associated with BC density, BC mass and BC
refractive index. Especially, the BC mass concentrations derived from AE33 measurements would
bear uncertainties associated with variations in mass absorption coefficient (Zhao et al., 2021b).
Sensitivity tests about these parameters are also included in Fig.S5. Results of previous study reveal
the refractive index of BC has almost minimal impact on scattering calculations (Ma et al., 2012a).
This is confirmed by the sensitivity test results shown in Fig. S5, with variations in possible reported
ranges of BC refractive index induce variations less than 2%. Results also show that uncertainties in
BC mass concentrations and BC density would only induce small scattering changes (near or less than

1%).

Note that, the sensitivity results shown here is somehow contrary to the conclusion draw by

Zhao et al. (2019b) that BC mass size distributions should have comparable impacts with BC mixing
states on simulations of aerosol scattering. This was further explored in Sect. S5. The results
demonstrate that the simulations in  Zhao et al. (2019b) assumes all aerosols contain BC, which is
not the case in ambient atmosphere, details about this discussion could be found in Sect. S5.

In this region, both primary and secondary aerosols contribute to aerosol absorption (Yuan et al.,

2016;Luo et al., 2022), which affect imaginary refractive index part ($m_i$) of BC-free aerosols. The
sensitivity tests about impacts of $m_i$ on scattering calculations were also included in Fig.S5. It shows
that varying $m_i$ from $10^{-2}$ to $10^{-7}$ (reported $m_i$ for different types of brown carbons (Saleh, 2020)) could
result in scattering changes of ~5%. However, even for biomass burning organic aerosols which are
the most absorbing aerosol type, their $m_i$ at 525 nm is on the order of $10^{-2}$. The BC-free aerosols during
this campaign mostly consist of inorganic and secondary organic aerosols. The overall $m_i$ of BC-free
aerosols are less than $10^{-3}$ estimated using the brown carbon absorption at 520 nm observed during the
campaign even when biomass burning activities prevail (Luo et al., 2022), meaning that scattering
deviations induced by errors $m_i$ in  assumption are less than 1% for assuming $m_i$ of $10^{-7}$.

Truncation errors and light source non-idealities were accounted for in the Mie calculations by

applying angular light intensity correction functions from Müller et al. (2011). Details of the Mie
theory calculations for the optical closure can be found in Sect. S4 of the supplement. Sensitivity tests





(discussed in Sect. 5 of the supplement) and above-mentioned discussions make it clear that $m_r$ is the
most influential parameter affecting the variations in aerosol scattering calculations.
Therefore, the optical closure calculations could be conducted iteratively to retrieve an $m_r$ value
that align simulated aerosol scattering at 525 nm with the measured scattering at the same wavelength
(Sect. S4). This retrieved $m_r$ at 525 nm, obtained through optical closure, is termed $m_{rc525}$. The
sensitivity tests results shown in Fig.S7 demonstrate the accuracy of $\sigma_{sp,PM_1}$ conversion from
represents one of the most important factors that would influence the accuracy of retrieved $m_{rc525}$.
The accuracy of $\sigma_{sp,PM_1}$ depends largely on the accuracy of ratio used for converting measured $\sigma_{sp,TSP}$
to $\sigma_{sp,PM_1}$ as discussed. Large standard deviation of the ratio $\sigma_{sp,TSP}/\sigma_{sp,PM_1}$ (>10%) exist for data
points of $\sigma_{sp,TSP}$ at 525 nm below 50 Mm⁻¹ (shown in Fig. S4b). For these points, even if correcting
the inlet inconsistency with the average curve shown in Fig.S4, large uncertainty would inevitably be
introduced to the optical closure. Therefore, $m_{rc525}$ was only retrieved when $\sigma_{sp,TSP}$ at 525 nm
exceeded 50 Mm⁻¹ (~75% of data points), where the ratio between $\sigma_{sp,PM_1}$ and $\sigma_{sp,TSP}$ varied with
standard deviations less than 5% (Fig.S4).

**3 Results and discussions**
**3.1 Strong Diameter Dependence of $m_{r1064}$ and Remarkable Difference in $m_{r,POA}$ and $m_{r,SOA}$**
**Revealed by Direct $m_{r1064}$ Measurements**
During the first campaign, significant variations in the $m_{r1064}$ were revealed using DMA-SP2
measurements (Fig. S9), with $m_{r1064}$ values ranging from 1.40 to 1.59 (mean: 1.49 ± 0.03). Fig.1a and
1b illustrate the diameter-dependent characteristics of the measured $m_{r1064}$. Aerosols larger than 300
nm generally exhibited higher $m_{r1064}$ values compared to those smaller than 300 nm, with average
$m_{r1064}$ values at diameters of 253 nm, 270 nm, 300 nm, 335 nm, 370 nm, and 400 nm being 1.46,
1.49, 1.48, 1.51, 1.51, and 1.51, respectively. This is consistent with previous findings indicating a
clear diameter dependence, where $m_{r1064}$ tends to increase with particle diameter (Benko et al.,
2009;Zhao et al., 2019a). Using the $m_{r1064}$ ratio between 400 nm and 235 nm ($m_{r1064,400}/m_{r1064,235}$)
as an indicator of $m_{r1064}$ diameter dependence, we found that this ratio increases with $m_{r1064,400}$ (Fig.
1b). Specifically, the ratio $m_{r1064,400}/m_{r1064,235}$ rose from 1.02 to 1.07 as $m_{r1064,400}$ increased from
1.46 to 1.58, while $m_{r1064,235}$ showed only a slight increase from 1.45 to 1.47. This suggests that the
chemical processes responsible for the increase in $m_{r1064,400}$ have minimal influence on the chemical
properties of aerosol particles near 235 nm, indicating that variations in $m_{r1064,400}$ and $m_{r1064,235}$ are



governed by different chemical and emission processes. The aerosol chemical compositions of NR-
PM$_1$ presented in Fig. 1b reveal that the condition corresponding to $m_{r1064,400}$ of 1.56 has an overall
higher content of MOOA. This is further confirmed by the probability distribution of MOOA mass
fractions in two regions of Fig.1b shown in Fig.S10. Although size distribution of secondary aerosols
matter, this general result suggests that secondary organic aerosol formation has possibly contributed
to the substantial increase in $m_{r1064,400}$. However, these results reflect bulk compositional changes
that may differ considerably from changes in composition fractions at 400 nm. Overall, these findings
indicate that aerosols with diameters near 235 nm and 400 nm likely originate from distinct sources,
and variations in $m_{r1064,235}$ and $m_{r1064,400}$ might reveal $m_r$ characteristics of different aerosols
sources.

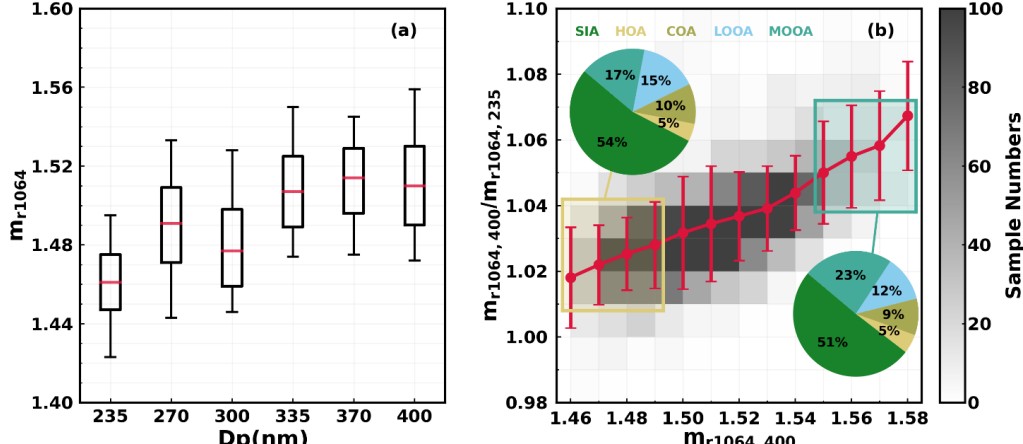

**Figure 1**. **(a)** The box-and-whisker plot (5th, 25th, 75th and 95th percentiles) of $m_{r1064}$ at different diameters; **(b)** Variations of $m_{r1064,400}/m_{r1064,235}$ under different $m_{r1064,400}$ levels, the intensity of colors indicating the numbers of samples, while red spots and error bars represent average values and standard deviations. Pie charts corresponding average aerosol compositions under different $m_{r1064,400}$ ranges.

As introduced in the previous section, the PMF source apportionment technique was applied to
aerosol chemical composition measurements and aerosol volume size distribution measurements
derived from PNSD measurements during the first short-term field campaign. Results of these two
approaches were combined to resolve the chemical fingerprints of sources at different diameters.
Fig.2a illustrates the average volume size distributions of five resolved factors based on PNSD
measurements. It shows that aerosols at a diameter of 235 nm are primarily contributed by factors 1,
3, and 5, with factor 3 being the most significant contributor. In contrast, aerosols at 400 nm are mainly
contributed by factors 1, 2, and 3, with factor 1 as the dominant contributor. As detailed in Sect. 2.2,
POA factors and SOA factors were resolved from the observed organic aerosol spectra. The average

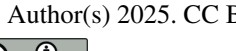


mass fractions during this campaign are as follows: POA (HOA + COA) at 15.4%, SOA (LOOA +
MOOA) at 34.7%, ammonium nitrate (AN) at 26.8%, and ammonium sulfate (AS) at 23.2%,
demonstrating a dominant contribution from secondary sources.

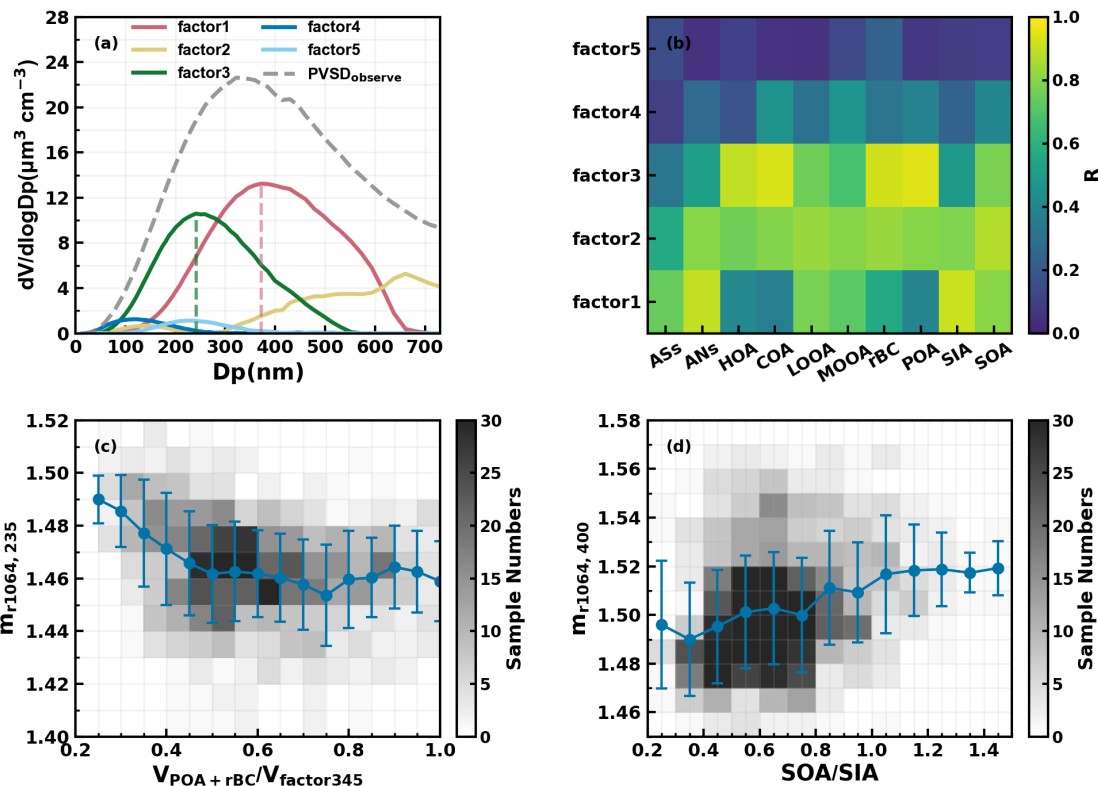

**Figure 2. (a)** The volume size distributions of factors from PNSD PMF analysis; **(b)** Correlation coefficient between volume concentrations of chemical compositions and resolved PNSD size factors; **(c)** $m_{r1064,235}$ varies with changes in the ratio of volume sum of POA and rBC to the volume sum of factor3, factor4 and factor5 (factor 345); **(d)** $m_{r1064,400}$ varies with SOA to SIA when mass summation of factor1 and factor 2 account for over 70% at 400 nm. The intensity of colors indicates the numbers of samples, while red spots and error bars represent average values and standard deviations in **(c)** and **(d)**.

Cai et al. (2020a) introduced an analytical method linking resolved aerosol size distributions to
different sources obtained from mass spectrometer measurements; this method is adopted here. The
correlation coefficients for mass concentrations of different aerosol sources and volume of different
factors are shown in Fig.2b. For factor 1, the volume peak size is around 400 nm, contributing an
average of 58% to the total measured aerosol volume, and shows a strong correlation with secondary
inorganic aerosols (SIA, R = 0.91), as well as a notable correlation with SOA (R = 0.80). Factor 2
contributes an average of 5.61% to the total volume and exhibits the highest correlation with SOA (R



= 0.87). The correlation analysis indicates that factors 1 and 2 are indeed linked to secondary sources,
with the correlation coefficient between the volume sums of factors 1 and 2 and the mass
concentrations of secondary components (SIA + SOA) reaching as high as 0.95. Taken together, these
findings suggest that secondary organic and inorganic particulate matter are dominant components in
particles larger than 400 nm, consistent with previous observations that larger particles are generally
more aged than smaller ones (Sun et al., 2012;Xu et al., 2021).
The volume size distribution of factor 3 ranges from 40 nm to 500 nm, with a geometric mean
diameter of approximately 240 nm, contributing 46% to the total volume concentration. It shows strong
correlations with HOA (R= 0.89), COA (R = 0.93), and rBC (R = 0.92). This finding aligns with
previous studies, which indicate that the peak volume size for HOA, COA, and rBC from traffic
emissions, typically occurs between 200 nm and 300 nm (Cai et al., 2020b;Sun et al., 2012;Xu et al.,
2021;Li et al., 2023a). Furthermore, the correlation coefficient between the volume of factor 3 and the
total mass concentrations of HOA, COA, and rBC reaches 0.95, suggesting that factor 3 is
predominantly associated with primary aerosol emissions. In addition, both factors 4 and 5 exhibit
smaller volume size distributions, collectively contributing 12% to the total volume in the 20 nm to
500 nm range. Although the volume contributions of factors 4 and 5 are generally minor, as shown in
Fig. 2a, their diurnal volume variations (illustrated in Fig. S3) and correlations with primary HOA,
COA, and rBC are consistently higher than their correlations with secondary species (Fig. 2b). This
suggests that factors 4 and 5 are more likely linked to primary sources.
Moreover, the total volume concentrations of factors 3, 4, and 5 are generally consistent with the
total volume derived from the mass concentrations of HOA, COA, and rBC (5.8 vs. 6.1 μm³/cm³).
However, in 53% of cases, the volume concentrations of factor 3 exceed those derived from the
measurements of HOA, COA, and rBC, indicating that factor 3 may sometimes include contributions
from other secondary sources, despite primary sources being dominant in most instances. Additionally,
as shown in Fig. 2a, while factors 3, 4, and 5 primarily contribute to the mass of aerosols with a
diameter of 235 nm, factor 1 also plays a significant role. This suggests that both primary emissions
and secondary aerosol formations influence variations in $m_{r1064,235}$. In contrast, variations in
$m_{r1064,400}$ are predominantly controlled by secondary aerosol sources, with primary emissions playing
a much lesser role. The differing sources of aerosols at diameters of 235 nm and 400 nm, along with
the distinct variation characteristics of $m_{r1064,235}$ and $m_{r1064,400}$, may provide insights into $m_r$
differences of POA and SOA.





Based on the PNSD PMF results, the volume contribution of factor 3 at a diameter of 235 nm
ranges from below 1% to nearly 99%, with an average of 60%. In comparison, the volume contribution
of factor 1 varies between nearly 0% and 93%, averaging around 30%. Fig.2c illustrates the variations
of $m_{r1064,235}$ as a function of the ratio between the total volume of POA and rBC and the total volume
of factors 3, 4, and 5 (referred to as factor 345) under conditions where volume of factor345 dominates
at 235 nm (volume fraction greater than 75%). The results indicate that as the contributions of POA
and rBC increase within factor 345, $m_{r1064,235}$ decreases from approximately 1.49 to about 1.46 when
their volume fraction exceeds 0.5, subsequently fluctuating within a narrow range ($1.46 \pm 0.02$). This
suggests that $m_{r1064,POA}$ is likely substantially lower than 1.46, considering that other secondary
species contribute more than 30% to aerosol volume at 235 nm under these conditions.
Both primary and secondary sources contribute to aerosols of 400 nm, however, volume fractions
of factor 1 plus 2 at 400 nm varied between near zero and 99% with an average of 61%, demonstrating
that aerosols at 400 nm composed primarily of SIA and SOA, thus variations in SOA and SIA mainly
control changes in $m_{r1064,400}$. With regard to $m_r$ values of AS and AN, $m_r$ of AS is consistently
reported as 1.53 (Tang, 1996;Stelson, 1990;Lide, 2004), while reported $m_r$ values of AN varied in a
relatively large range of 1.41 to 1.56 (Jarzembski et al., 2003;Lide, 2004;Ouimette and Flagan,
1982;Schuster et al., 2005;Zhang et al., 2012). However, known $m_r$ values of AS and AN demonstrate
that it is likely not the formation of SIA that has led to $m_{r,400}$ to be as high as 1.58, and could most
possibly result from SOA formations, and results of previous laboratory (Li et al., 2017) and field
studies (Zhao et al., 2021a) suggested that SOA formations might enhance $m_r$ to reach beyond 1.6.
However, the direct subtraction of $m_{r1064,SOA}$ with current measurements is quite challenging due to
that the lack of size distribution measurements of aerosol chemical compositions. To give a glimpse
into the influences of SOA, the variations of $m_{r1064,400}$ under different mass ratios of SOA to SIA are
shown in Fig.2d. It shows that on average, $m_{r1064,400}$ indeed increases as a function of SOA fraction,
confirming that the $m_{r1064,SOA}$ is higher than those of SIA. Especially, the $m_{r1064,400}$ showed
obviously higher correlations with MOOA (R=0.25) than with LOOA (R=-0.24) (Fig.S11),
demonstrating that the higher $m_r$ of MOOA than LOOA, and the $m_r$ of LOOA is likely close to those
of SIA on the basis of results shown in Figure.S8b.  Results of a few existing field measurements
supports the finding. For example, results of  Aldhaif et al. (2018) suggested that $m_{r,OA}$ were likely
positively correlated with the O/C.  Results of Liu et al. (2022) revealed much stronger scattering
abilities of MOOA which likely cannot be solely explained by larger size of MOOA. Li et al. (2023c)
established a semi-empirical model to predict the $m_{r,OA}$ from O/C and H/C which was partly





confirmed using their laboratory measurements. On the basis of this model and element ratios of OA
factors, $m_r$ values for retrieved HOA, COA, LOOA, and MOOA during the first campaign are 1.48,
1.48, 1.45, 1.63, respectively, which is partly consistent with the field finding of this research that
$m_{r,POA}$ has remarkable difference with $m_{r,SOA}$ with $m_r$ of MOOA is substantially higher, however,
the $m_r$ values for HOA, COA might be overestimated by the scheme of Li, et al. (2023c).
**3.2 Optical Closure Confirm Remarkably higher $m_{r,SOA}$ than $m_{r,POA}$**

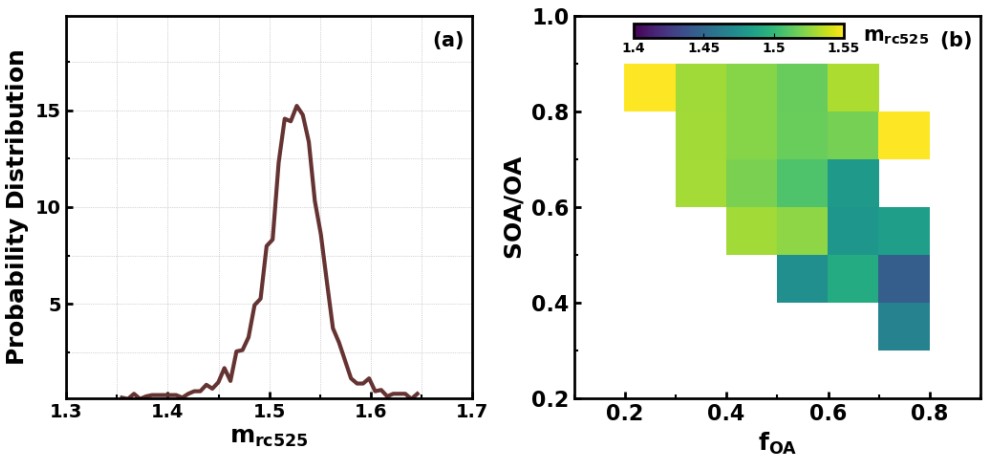

**Figure 3**. **(a)** Probability distribution of $m_{rc525}$; **(b)** Variations of $m_{rc525}$ under different OA mass fractions in NR-PM$_1$ (f$_{OA}$) and SOA mass fractions in total OA (SOA/OA).

The probability distributions of retrieved $m_{rc}$ using the optical closure method presented in Sect
2.3 is presented in Fig.3a, $m_{rc525}$ generally ranges between 1.4 and 1.6 with an average of 1.52, which
is close to the value of 1.53 typically used in optical closure studies on the basis of field measurements
(Ma et al., 2011;Wexler and Clegg, 2002), and also generally consistent with the previously reported
$m_r$ range in Guangzhou (Zhang et al., 2016a). The retrieved $m_{rc525}$ is a parameter that represents the
bulk $m_r$ of ambient aerosols, therefore, containing influences of both organic and inorganic
components. To further reveal possible effects of POA and SOA on $m_{rc525}$ variations, average $m_{rc525}$
under different OA mass fractions in NR-PM$_1$ and SOA fractions in OA are presented in Fig.3b.  A
general characteristic revealed that when OA accounts for more than 50% of NR-PM$_1$, $m_{rc525}$ tends
to be lower as mass fraction of POA increase, suggesting that increases in POA would generally lower
the $m_{rc525}$. The fact that $m_{rc525}$ would be even lower than 1.45 when POA dominates, suggesting that
$m_{rc525,POA}$ is likely lower than 1.45 which is consistent with the finding revealed in Sect 3.1 that
$m_{r1064,POA}$ should be lower than 1.46. Results of Liu et al. (2013) revealed that small $m_r$ wavelength
dependence of organic aerosols for wavelengths higher than 500 nm. The finding about $m_{rc525,POA}$





and $m_{r1064,POA}$ here demonstrates that both optical closure calculations and DMA-SP2 measurements
reveal same results on the value of $m_{r,POA}$. However, as SOA dominates in OA, the $m_{rc525}$ is
enhanced to more than the average of 1.52, suggesting that the $m_{rc525,SOA}$ is at least higher than 1.52
considering that $m_{r525}$ of AS and AN is close to or lower than 1.52. These results qualitatively
confirmed the finding in Sect 3.1 that $m_{r,SOA}$ is substantially higher than $m_{r,POA}$.

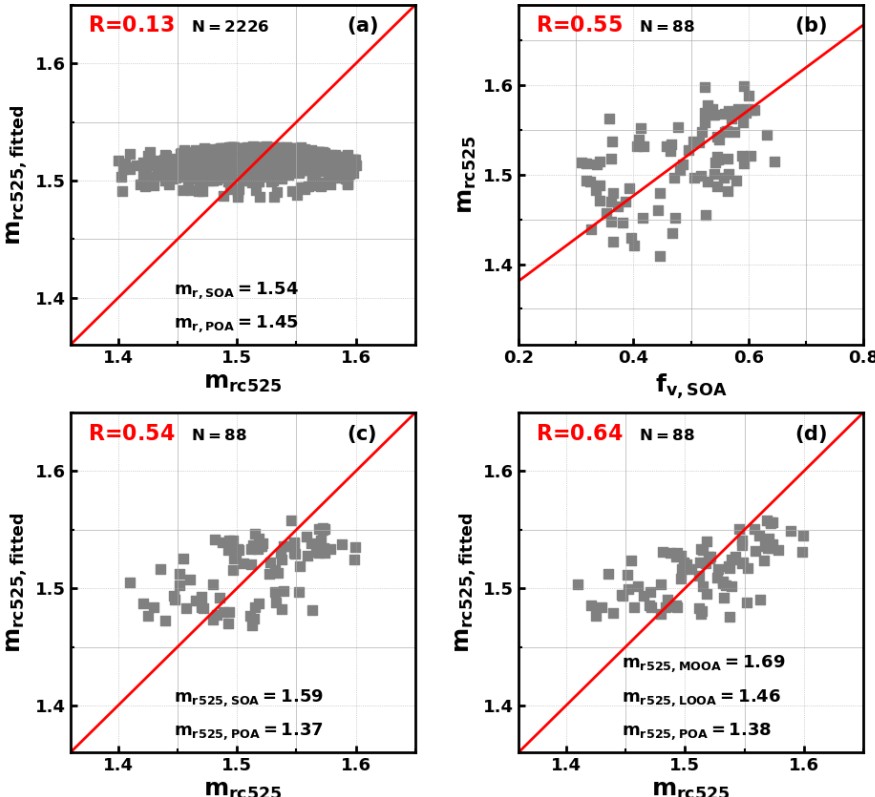

**Figure 4**. **(a)**Comparison between $m_{rc525}$ and fitted $m_{r525}$ using the volume mixing rule, N is the sample number; **(b)** Relationships between $m_{rc525}$ and volume fractions of SOA in NR-PM$_1$ (f$_{v,SOA}$) when OA volume fractions in NR-PM$_1$ is higher than 75%; **(c)**Comparisons between $m_{rc525}$ and fitted $m_{r525}$ for points in (b) using the volume mixing rule; **(d)** Comparisons between $m_{rc525}$ and fitted $m_{r525}$ for points in (b) using the volume mixing rule, while LOOA and MOOA are treated separately.

The complex compositions of ambient aerosols and complex mixing states and refractive index
mixing rule (Liu and Daum, 2008;Zhao et al., 2019a) result in the fact that retrieving $m_{r,SOA}$ and





$m_{r,POA}$ on the basis of direct $m_{rc525}$ measurements and aerosol composition measurements is
extremely challenging (Zhao et al., 2022). Actually, $m_{r,SOA}$ and $m_{r,POA}$ are never reported on the basis
of field measurements.  Volume mixing rule was first applied in this study to all retrieved $m_{rc525}$
points and corresponding aerosol chemical compositions to retrieve $m_{rc525,SOA}$ and $m_{rc525,POA}$. The
multilinear regression formula with the volume mixing rule could be expressed as $m_{r525}=\varepsilon_{AN} \times$
$m_{rc525,AN} + \varepsilon_{AS} \times m_{rc525,AS} + \varepsilon_{POA} \times m_{rc525,POA} + \varepsilon_{SOA} \times m_{rc525,SOA} + \varepsilon_{AC} \times m_{rc525,AC}$, where $\varepsilon$
represents volume fraction of each species in total measured NR-PM$_1$, $\varepsilon_{AC}$ represents volume fraction
of ammonium chloride by assuming all measured chloride mass is ammonium chloride. In this formula,
the $m_{rc525,AS}$ and $m_{rc525,AC}$ is set to 1.53 and $m_{rc525,AN}$ is set to 1.5 (a middle value reported in the
literatures) in the retrieval in terms of existing literatures as listed in Sect 3.1. As shown in Fig.4a, if
all retrieved $m_{rc525}$ points were used,$m_{rc525,POA}$ and $m_{rc525,SOA}$ of 1.45 and 1.54 could be retrieved
(using python curve_fit function of module scipy.optimize). However, it could be found that this rule
could not reproduce observed large variations in $m_{rc525}$. It was realized that the volume mixing rule
would oversimplify the interactions in complex mixture with respect to aerosol refractive index (Liu
and Daum, 2008). This was confirmed by a few aerosol refractive index studies (Zhao et al., 2019a).

If we focus on periods when OA dominates aerosol volume (OA volume in total NR-PM$_1$

accounts for more than 75%), $m_{rc525}$ shows clear almost linear trend with the volume fractions of
SOA in NR-PM$_1$ as shown in Fig.4b. The question remains whether the volume mixing rule can be
used to retrieve $m_{rc525}$ of POA and SOA from these organic aerosol-dominant points. The analysis
in Sect. 3.1 demonstrates that the mass concentration of resolved PNSD factor that related with primary
sources could mostly be explained by POA and BC, meaning that PNSD and of primary sources and
secondary sources could be generally separated in the PMF procedure, suggesting that POA and SOA
are likely prone to externally mixed. This could be explained by that the observation site is located
near POA source regions in an urban area. This indicates that POA and SOA tend to be optically
independent at the single-particle level. The optical test experiment introduced in Sect. S6 demonstrate
that the volume mixing rule can generally be used to retrieve $m_{rc525}$ of POA and SOA if aerosol
particles of SOA and POA are externally mixed. In view of this, the volume mixing rule was further
applied to scenarios when OA volume dominates (data points in Fig.4b), meaning that effects of other
chemical components are minimized, the overall $m_{rc525}$ changes were captured by the fitting while
retrieved values for  $m_{rc525,POA}$ and $m_{rc525,SOA}$ are $1.37 \pm 0.026$ and $1.59 \pm 0.017$ as shown in Fig.4c.
The relatively small uncertainty ranges for estimated $m_{rc525,POA}$ and $m_{rc525,SOA}$ confirms the
quantitative difference between $m_{rc525,POA}$ and $m_{rc525,SOA}$. For the average PNSD in the retrievals,





increase $m_{rc525}$ from 1.37 to 1.59 would in an ~60% increase of aerosol scattering, which is significant
for aerosol radiative forcing estimations (Kuang et al., 2015). Models that assume $m_{r,POA}$ to 1.63 and
$m_{r,SOA}$ to 1.43 which is contrary to the finding here, would inevitably results significant bias in organic
aerosol radiative forcing estimations (Curci et al., 2019). Based on the above analysis, $m_{r,SOA}$ and
$m_{r,POA}$ of 1.37 and 1.59 might be better choice for model settings.

Considering that O/C of LOOA and MOOA differ much (0.6 vs 1.27), if SOA was further treated

separately as LOOA and MOOA in the fitting, extending the volume mixing rule formula as:
$m_{r525} = \varepsilon_{AN} \times m_{rc525,AN} + \varepsilon_{AS} \times m_{rc525,AS} + \varepsilon_{AC} \times m_{rc525,AC} + \varepsilon_{POA} \times m_{rc525,POA} + \varepsilon_{LOOA} \times$
$m_{rc525,LOOA} + \varepsilon_{MOOA} \times m_{rc525,MOOA}$, a better correlation coefficient could be achieved as shown in
Fig.4d. Note that LOOA and MOOA are generally not externally mixed and likely be prone to
internally mixed on the basis of knowledge about organic aerosol aging chain (Jimenez et al., 2009),
therefore the volume mixing rule is likely not applicable, and the retrieved results serves better for
qualitatively   analysis.   Retrieved   $m_{rc525,POA}$ of  1.38 ± 0.024,   and   retrieved   $m_{rc525,LOOA}$ and
$m_{rc525,MOOA}$ of 1.46 ± 0.069 and 1.69 ± 0.059, consistent with the speculations in Sect 3.2 that
$m_{r,MOOA}$ is likely substantially higher than $m_{r,LOOA}$ (Fig.S11), although significant retrieval bias of
$m_{rc525,LOOA}$ and $m_{rc525,MOOA}$. The retrieved $m_{rc525,LOOA}$ and $m_{rc525,MOOA}$ of 1.46 and 1.69 have a
remarkable difference with those ($m_{r,LOOA}$ and $m_{r,MOOA}$ of 1.56 and 1.57) predicted with their O/C
and H/C ratios as inputs of the scheme proposed by Li, et al. (2023c). This result suggests that
qualitatively, $m_r$ increase with oxidation degree of SOA,  which is contrary to results of most existing
laboratory studies that increase of O/C would decrease $m_r$ at the O/C range of LOOA and MOOA (He
et al., 2018;Moise et al., 2015).  This is likely associated with that MOOA in Guangzhou urban area
mainly formed through multiphase reactions (Zhai et al., 2023) thus has higher $m_r$ as demonstrated by
Li et al. (2017) that multiphase reactions enhance $m_r$, while most laboratory studies on evolution of
$m_{r,SOA}$ were conducted in the context of gas-phase reactions.
**4 Conclusions**

This study thoroughly leverages field measurements and multiple analytical techniques to

constrain the real part of the scattering refractive index of organic aerosols. The results reveal
substantially higher values for SOA compared to POA, helping to clarify a longstanding discrepancy
in their optical properties. The $m_r$ is a fundamental parameter for accurate simulations of aerosol
optical properties and their roles in visibility degradation and direct aerosol radiative forcing. In
addition, aerosol optical properties are also key for radiative flux simulations which are fundamental



for atmospheric photochemistry (Tao et al., 2014;Tian et al., 2019). Therefore, results of studies have
both significant implications in environmental and climate issues.

Results of Redemann et al. (2000) demonstrate that 5% variation in $m_r$ can lead to approximately

30% change in the radiative flux change at the top of atmosphere. Li et al. (2021) further demonstrated
for $m_{r,OA}$ changing from 1.3 to 1.65, stratospheric aerosol optical depth relatively changed from −20%
to +50%, and caused up to ±100% variability in shortwave radiative forcing, which matters more than
mixing state. While OA are mainly composed of POA and SOA, and they both are major components
of atmospheric aerosols, therefore accurate representations of $m_{r,POA}$ and $m_{r,SOA}$ are essential for
accurate simulations of direct aerosol radiative effects whose uncertainties are the second largest
contributions to overall climate forcing estimations (IPCC,AR6, 2023). Our long-term field
observation results suggest that utilizing constant values for $m_{r,OA}$ in models would lead to either
significant underestimations or overestimations in scattering coefficient therefore significant
deviations in estimations of direct aerosol radiative effects. The used constant value is another issue,
as presented by Tsigaridis and Kanakidou (2018), most models use 1.53 as $m_{r,OA}$ which is generally
appropriate on the basis of this study if $m_{r,OA}$ has to be assumed. While some models even use a
constant, for example 1.45 for both SOA and POA which might cause systematical underestimation
of OA scattering (Aouizerats et al., 2010;Ma et al., 2012b). If OA is further categorized into SOA and
POA in models as applied in Curci, et al. (2019), the appropriate $m_{r,POA}$ and $m_{r,SOA}$ should be used.
Large bias would be expected if $m_{r,POA}$ and $m_{r,SOA}$ are set to 1.63 and 1.43 as those in Curci, et al.
(2019). In addition, in most models, element ratios of organic components are not available, and
organic aerosols are generally categorized as several types of POA and SOA, and SOA are generally
treated as a whole in these models (Zhang et al., 2023;Pöhlker et al., 2023). Therefore, $m_{r,POA}$ and
$m_{r,SOA}$ values of 1.37 and 1.59 retrieved at 525 nm in this study are recommended for model settings.

It should be noteworthy that the POA in this study are primarily composed of fossil combustion

and cooking related organic aerosols, however, organic aerosols directly emitted from biomass burning
(BBOA) also represent a major POA source. Mathai et al. (2023) reported $m_r$ of homogeneously and
inhomogeneously mixed tar balls in the free troposphere from biomass burnings as 1.26 and 1.4, which
is close to the recommended $m_{r,POA}$. However, results of Womack et al. (2021) reported that $m_r$ of
biomass burning aerosols at 475 nm could be higher than 1.6. Results of Luo et al. (2022) further
demonstrated that $m_r$ of BBOA might vary a lot and depends highly on combustion conditions.
Accurate representations of $m_{r,BBOA}$ stand as urgent need, considering the increasing trends of
biomass burning events under background of current global warming. However, the biggest challenge





lies in accurate representations of organic aerosols $m_r$ due to $m_{r,SOA}$ variations, because SOA could
be formed through varying pathways of different precursor sources volatile organic compounds
(biogenic versus anthropogenic), and existing results already proved that SOA formed from varying
precursors and pathways has distinct $m_r$. Therefore, recommend value of $m_{r,SOA}$ in this study might
represent more the $m_r$ of SOA that formed from anthropogenic precursors in urban regions. Overall,
results of this study, underscore the substantially higher $m_{r,SOA}$ than $m_{r,POA}$, not the case currently
assumed in models.
In addition, results of this study imply that $m_r$ likely increased oxidation level, suggesting crucial
impacts of SOA formation mechanisms on $m_r$ variations. Future studies should further examine
variations and evolution of   $m_{r,SOA}$ than $m_{r,POA}$ under different emissions characteristics and
chemistry pathways for reducing uncertainties of direct aerosol radiative effects simulations in
chemical, weather and climate models.


**Competing interests**. The contact author has declared that none of the authors has any competing
interests.


**Author Contributions.** YK and LL designed the two field campaigns, YK conceived and led this
research. LL, BL, JLS, HX, GZ, FLY, MMZ, FL and YK performed measurements of aerosol physical
and chemical properties. TD and XD supported this campaign. JLS performed the analysis with YK
and LL, JLS and YK wrote the manuscript. All authors contributed to revisions of this paper.


**Financial support**. This work is supported by National Natural Science Foundation of China
(42175083, 42105092), the Fundamental Research Funds for the Central Universities.



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
