# Peer review of "Refractive Index Enhancement by Secondary Organic Aerosol"

_EGUsphere, 2025_

## Referee Comment (RC1)

Review of "Field Observations Reveal Substantially Higher Scattering Refractive Index in Secondary Versus Primary Organic Aerosols" by Shen et al.

This work uses field measurements of aerosol size, composition, and optical properties to constrain estimates of the real part of the refractive index for organic aerosol. Based on statistical characterization of the organic spectra using PMF, the authors were able to identify organic aerosol types as having primary and secondary sources. Combined with this identification and direct DMA-SP2 measurements, the authors quantified contributions of the different aerosol types to the refractive index and recommended revised estimates of refractive index for POA and SOA. The analysis in this paper is cleverly carried out and thorough, providing fidelity to the arguments and new refractive index recommendations. The figures clearly illustrate discussions in the text. The paper is generally well written but there are a lot of grammatical errors and the sentence structure is at times difficult to follow. In some places I noted where these were and made suggestions, but the issues were far too numerous to point out each one. The authors should make a concerted effort to carefully re-read the paper to ensure its clarity. I believe this paper is appropriate for this journal but should only be accepted after the authors address the following minor comments.

1. Line 109-110: The authors should cite the manufacturer of the SP2 (Droplet Measurement Technologies).

2. Lines 105-115: The authors have not provided any discussion on the set up or quantified the inlet and loss properties of the instrument set up and sampling system. This information must be included. What altitude were the aerosol sampled from? Were the measurements continuous? Were the aerosol collected in a container using one main inlet? Was there an impactor or cyclone before sampling into the instruments? What was the main inlet flow and instrument flows? Have the authors quantified the sampling losses to the instruments? How were the aerosol dried? How was the relative humidity monitored?

3. Line 116: can the authors provide a citation for how this equation was derived?

4. Line 140: what type of nephelometer was used? Please provide the same information where relevant as in my second comment.

5. Line 166: no need to redefine PMF.

6. Figure S4 caption: revise text to, "[Comparison of] dry-state aerosol …"

7. Figure S5 and S7: how is the relative deviation calculated? Please specify.

8.  Line 302-304: The authors state that the chemical composition at $mr1064,400$ of 1.56 has higher MOOA content. Higher than what? The $mr1064,400$ at 1.48?

Please specify which mr1064,400 and MOOA content are being referenced in the text.

9. Figure 4: please specify that the red line in these panels is the 1:1 line in the caption.

10. Line 454: Grammar: "…increase mrc525 from 1.37 to 1.59 would [result?] in a ~60%..."

11. Line 474-477: This sentence is too long and confusion. It should be split into two sentences for clarity.

12. Data availability: In accordance with ACP guidelines (https://www.atmospheric-chemistry-and-physics.net/policies/data_policy.html) that "authors are required to provide a statement on how their underlying research data can be accessed", the authors must provide a resource that contains their deposited data to "guarantee the integrity, transparency, reuse, and reproducibility of scientific findings." If this is not possible, the authors need to clarify why their data is not being shared. This should be in the Data Availability section at the end of the manuscript.

---

## Author Comment (AC1)

**Responses to anonymous referee #1**

**General comment**:

This work uses field measurements of aerosol size, composition, and optical properties to constrain estimates of the real part of the refractive index for organic aerosol. Based on statistical characterization of the organic spectra using PMF, the authors were able to identify organic aerosol types as having primary and secondary sources. Combined with this identification and direct DMA-SP2 measurements, the authors quantified contributions of the different aerosol types to the refractive index and recommended revised estimates of refractive index for POA and SOA. The analysis in this paper is cleverly carried out and thorough, providing fidelity to the arguments and new refractive index recommendations. The figures clearly illustrate discussions in the text. The paper is generally well written but there are a lot of grammatical errors and the sentence structure is at times difficult to follow. In some places I noted where these were and made suggestions, but the issues were far too numerous to point out each one. The authors should make a concerted effort to carefully re-read the paper to ensure its clarity. I believe this paper is appropriate for this journal but should only be accepted after the authors address the following minor comments.

**Response**: Thanks for your comments, which really helped improve the manuscript and we have scrutinized the manuscript to ensure its clarity.

**Minor Comments**:

**Comment:** Line 109-110: The authors should cite the manufacturer of the SP2 (Droplet

Measurement Technologies).

**Response**: The manufacturer cited as "with single-particle soot photometer from Droplet Measurement Technologies, Boulder, Colorado (Schwarz et al., 2006)."

**Comment**: Lines 105-115: The authors have not provided any discussion on the set up or quantified the inlet and loss properties of the instrument set up and sampling system. This information must be included. What altitude were the aerosol sampled from? Were the measurements continuous? Were the aerosol collected in a container using one main inlet? Was there an impactor or cyclone before sampling into the instruments? What was the main inlet flow and instrument flows? Have the authors quantified the sampling losses to the instruments? How were the aerosol dried? How was the relative humidity monitored?

**Response**: Thanks for pointing this out. This information was added as "Aerosol absorptions at multiple wavelengths were measured using the AE33 from MAGEE (Drinovec et al., 2015). Note that a PM2.5 inlet (BGI, SCC 2.354) with a required flow rate of 8 L/min was used for aerosol sampling, with a Nafion drier of 1.2 m length downstream of the impactor, which ensures the sampling RH in instruments could be down to around 10% as recorded by the inlet RH sensor of the Q-ACSM. The flow rates of the Q-ACSM, SMPS, SP2 and AE33 instruments were set to 3 L/min, 0.3 L/min, 0.1 L/min, and 5 L/min, respectively. Nafion drier and all sampling tudes before instruments were placed vertically to reduce sampling loss."

**Comment**: Line 116: can the authors provide a citation for how this equation was

derived?

**Response**: The reference was added: (Zhao et al., 2019)

**Comment**: Line 140: what type of nephelometer was used? Please provide the same information where relevant as in my second comment.

**Response**: Added, "using a nephelometer (Aurora 3000 from Ecotech, (Müller et al., 2011))"

**Comment**: Line 166: no need to redefine PMF.

**Response**: Revised.

**Comment**: Figure S4 caption: revise text to, "[Comparison of] dry-state aerosol …".

**Response**: Revised.

**Comment**: Figure S5 and S7: how is the relative deviation calculated? Please specify.

**Response**: Clarified, "The ranges of relative deviations were calculated as the relative differences in scattering coefficients between simulations using the lowest and highest values of the variable and those using the default value represented as black circle."

**Comment**: Line 302-304: The authors state that the chemical composition at mr1064,400 of 1.56 has higher MOOA content. Higher than what? The mr1064,400 at 1.48? Please specify which mr1064,400 and MOOA content are being referenced in the text.

**Response**: Revised as "corresponding to $m_{r1064,400}$ of 1.56 has an overall higher content of MOOA than that near $m_{r1064,400}$ of 1.48."

**Comment**: Figure 4: please specify that the red line in these panels is the 1:1 line in the caption.

**Response**: Added, "Red lines indicate the 1:1 reference lines."

**Comment**: F Line 454: Grammar: "···increase mrc525 from 1.37 to 1.59 would [result?] in a ~60%..."

**Response**: Yes, result in, revised.

**Comment**: Line 474-477: This sentence is too long and confusion. It should be split into two sentences for clarity.

**Response**: Split into two sentences: "This is likely associated with that MOOA in Guangzhou urban area is mainly formed through multiphase reactions (Zhai et al., 2023) thus has higher $m_r$ as demonstrated by Li et al. (2017) that multiphase reactions enhance $m_r$. Most laboratory studies on evolution of $m_{r,SOA}$ were conducted in the context of gas-phase reactions."

**Comment**: Data availability: In accordance with ACP guidelines (https://www.atmospheric chemistry-and-physics.net/policies/data_policy.html) that "authors are required to provide a statement on how their underlying research data can

be accessed", the authors must provide a resource that contains their deposited data to "guarantee the integrity, transparency, reuse, and reproducibility of scientific findings." If this is not possible, the authors need to clarify why their data is not being shared. This should be in the Data Availability section at the end of the manuscript.

**Response**: The following statement was added "Data Availability. All data presented in Figures of this manuscript are freely available at https://doi.org/10.5281/zenodo.15786937, and more specific raw data will be made available on request due to the data restriction policy."

**References:**

Drinovec, L., Močnik, G., Zotter, P., Prévôt, A. S. H., Ruckstuhl, C., Coz, E., Rupakheti, M., Sciare, J., Müller, T., Wiedensohler, A., and Hansen, A. D. A.: The "dual-spot" Aethalometer: an improved measurement of aerosol black carbon with real-time loading compensation, Atmospheric Measurement Techniques, 8, 1965-1979, 10.5194/amt-8-1965-2015, 2015.

Li, K., Li, J., Liggio, J., Wang, W., Ge, M., Liu, Q., Guo, Y., Tong, S., Li, J., Peng, C., Jing, B., Wang, D., and Fu, P.: Enhanced Light Scattering of Secondary Organic Aerosols by Multiphase Reactions, Environmental science & technology, 51, 1285-1292, 10.1021/acs.est.6b03229, 2017.

Müller, T., Laborde, M., Kassell, G., and Wiedensohler, A.: Design and performance of a three-wavelength LED-based total scatter and backscatter integrating nephelometer, Atmos. Meas. Tech., 4, 1291-1303, 10.5194/amt-4-1291-2011, 2011.

Schwarz, J. P., Gao, R. S., Fahey, D. W., Thomson, D. S., Watts, L. A., Wilson, J. C., Reeves, J. M., Darbeheshti, M., Baumgardner, D. G., Kok, G. L., Chung, S. H., Schulz, M., Hendricks, J., Lauer, A., Kärcher, B., Slowik, J. G., Rosenlof, K. H., Thompson, T. L., Langford, A. O., Loewenstein, M., and Aikin, K. C.: Single-particle measurements of midlatitude black carbon and light-scattering aerosols from the boundary layer to the lower stratosphere, Journal of Geophysical Research: Atmospheres, 111, D16207, 10.1029/2006JD007076, 2006.

Zhai, M., Kuang, Y., Liu, L., He, Y., Luo, B., Xu, W., Tao, J., Zou, Y., Li, F., Yin, C., Li, C., Xu, H., and Deng, X.: Insights into characteristics and formation mechanisms of secondary organic aerosols in the Guangzhou urban area, Atmos. Chem. Phys., 23, 5119-5133, 10.5194/acp-23-5119-2023, 2023.

Zhao, G., Zhao, W., and Zhao, C.: Method to measure the size-resolved real part of aerosol refractive index using differential mobility analyzer in tandem with single-particle soot photometer, Atmos. Meas. Tech., 12, 3541-3550, 10.5194/amt-12-3541-2019, 2019.

---

## Author Comment (AC2)

**Responses to anonymous referee #2**

**General comment**:

While most researchers nowadays focus on light absorption by brown carbon when trying to address the climate effect of aerosols, Shen et al. focused on the overlooked aerosols scattering, which is simplified in a lot of studies, that may induce large uncertainty in radiative forcing estimation. The topic fits well with the scope of ACP. The manuscript is generally well written and adequately organized. I have a few comments before it can be accepted for further consideration.

**Response**: Thanks for your comments, which really helped improve the manuscript.

**Comment:** In the abstract and lines 471-474. The authors found an increasing trend of real refractive index with O/C or aging and stated this finding conflicts with most lab results. For example, He et al. (2018) observed a declining trend of real refractive index when OA aged from LO-OOA to MO-OOA for photooxidation of b-pinene and p-xylene. However, He et al (DOI: 10.1021/acs.est.1c07328) and Li et al. (DOI: 10.5194/acp-19-139-2019) also observed increasing refractive index with aging for naphthalene+NOx and biomass burning aerosols. Lab experiments did not conclude on this aspect, as it seems to depend on what kind of aerosol was studied. I would say such of statement exaggerates the significance of the finding in this study.

**Response**: Thank you for pointing this out. We had not previously noted the findings of these two studies. We agree with the reviewer that the relationship between $m_r$ and O/C can vary depending on aerosol types and oxidation conditions. However, as summarized

in Moise et al. (2015), within the O/C range relevant to the LOOA and MOOA identified in our study, most laboratory results show a decreasing trend in $m_r$ with increasing O/C. We have revised the discussion in this section accordingly. This aspect represents only a minor part of our study, and we did not intend to overstate its importance. Our goal is to interpret our results in the context of the broader literature.

"This result suggests that qualitatively, $m_r$ increase with oxidation degree of SOA. Previous laboratory results demonstrate that $m_r$ could increase (Li et al., 2019;He et al., 2022) or decrease (He et al., 2018) with O/C depending on aerosol types, precursors and oxidation conditions. However, as concluded in Moise et al. (2015) , most existing laboratory studies reveal that increase of O/C would decrease $m_r$, especially at the O/C range of LOOA and MOOA of this study (0.6 to 1.27, and $m_r$ from 1.46 to 1.69). This is likely associated with that MOOA in Guangzhou urban area is mainly formed through multiphase reactions (Zhai et al., 2023) thus has higher $m_r$ as demonstrated by Li et al. (2017) that multiphase reactions enhance $m_r$. Most laboratory studies on evolution of $m_{r,SOA}$ were conducted in the context of gas-phase reactions."

We also revised the sentence in the abstract as:

"In addition, further analysis reveals m_r of SOA increases substantially with oxidation level which is likely associated with multiphase SOA formation."

**Comment**: The results obtained from this study are from a single observation site near the emission source. How could the results and conclusions from this unique location be applied to a larger scale or different locations with potentially different emission

sources and experiencing different atmospheric processes?

**Response**: We agree the reviewer that observation at this single observation site might not be applied in regions does not share similar emissions and meteorological conditions with the observation site. The most exciting part of this study for us is the comparisons between scattering refractive index of POA and SOA could be directly revealed, and the resulted have challenged current model settings. We revised the recommendation in the abstract.

"Our analysis recommends $m_r$ values at 525 nm of 1.37 for POA and 1.59 for SOA in urban regions which share similar emissions and meteorological conditions as the observation site."

The following discussions were added in the conclusions

"However, the biggest challenge lies in accurate representations of organic aerosols $m_r$ due to $m_{r,SOA}$ variations, because SOA could be formed through varying pathways (condensational growth or multiphase reactions) of different precursor sources volatile organic compounds (biogenic versus anthropogenic), and existing results already proved that SOA formed from varying precursors and pathways has distinct $m_r$. Therefore, recommend value of $m_{r,SOA}$ in this study might represent more the $m_r$ of SOA that formed from anthropogenic precursors in urban and humid regions. "

The related sentence in the discussions was modified accordingly

"Based on the above analysis, $m_{r,SOA}$ and $m_{r,POA}$ of 1.37 and 1.59 might be better choice for model settings in regions share similar emissions and meteorological conditions with the observation site. "

Also, we have also revised the title as "Secondary Organic Aerosol Formation Enhances Refractive Index in Humid Southern China Challenging Model Assumptions" to emphasizes both the process, the location, and the implication (model challenge) in a balanced way.

**Comment**: I am not clear how one could use PMF to do source apportionment for PNSD. The authors also mentioned in their manuscript that LOOA and MOOA are generally not externally mixed and are likely to be prone to internal mixing (lines 463-465). But in the discussion of lines 440-446, the authors stated that POA and SOA tend to be optically independent at the single-particle level. The authors did observe SOA at the observation site. According to our knowledge of new particle formation and condensation growth, SOA and POA would not be completely externally mixed. Condensation of secondary products on existing particles (e.g., POA) would change the size of the particles, but does not change the number. How could PMF deal with this situation?

**Response**: We agree with the reviewer that SOA and POA are not completely independent. However, their degree of interaction largely depends on the time elapsed since POA emission and the extent to which SOA vapors have condensed onto POA particles. The organic aerosol (OA) observed at a given site typically comprises freshly emitted POA, preexisting SOA transported within the air mass, slightly aged POA (which may sometimes be classified as OOA and thus as SOA), and newly formed,

locally produced SOA.

Because our observation site is close to emission sources, a significant fraction of the measured OA signal likely originates from fresh and slightly aged POA. If POA particles differ in size from preexisting aerosols, they can be distinguished using the PMF procedure, as demonstrated by Cai et al. (2020). In our study, PMF analysis revealed that POA exhibited a distinct size distribution from secondary aerosols, with POA mass peaking around 250 nm, whereas the mass of secondary components peaked near 400 nm. A fully internally mixed state would be expected to yield identical mass size distributions for POA and SOA, which is not supported by our observations. Our observations support that POA and SOA reside in distinct size ranges. This indicates that although some internal mixing of POA and SOA may occur, the majority of their mass is likely externally mixed, as they predominantly reside in distinct size modes.

**Comment**: Figure 1b, for me, it looks like the mr1064,400/mr1064,235 changed suddenly at mr1064,400=1.53. What is the mechanism behind this?

**Response**: The ratio mr1064,400/mr1064,235 does not exhibit a sudden change at 1.53; instead, it generally increases smoothly with mr1064,400. The number of data points significantly decreases for mr1064,400>1.53, likely because inorganic aerosol components dominate over secondary organic aerosols in most cases, making instances where mr1064,400 exceed 1.53 relatively rare.

**Comment**: Line 386-387, would the correlation coefficient of R=0.25 and R=-0.24 be significantly different?

**Response**: Whether this difference is statistically significant depends strongly on the sample size. Given that the sample size exceeds 500 and the trend is clearly visible in Fig. S11, the difference is considered statistically significant.

**Comment**: The authors used the empirical method proposed by Li et al. (2023). I would say this might not be so relevant for this paper. Li's method was developed based on refractive index data for pure compounds without N element. However, in urban locations, aerosols already have nitrogen-containing species that affect the refractive index of the aerosols. The explanations in lines 474-478 are not well supported, as the aerosols themselves are different. It is not clear how much the formation pathway matters.

**Response**: We agree with the reviewer that the method proposed by Li et al. does not include nitrogen (N) elements. However, their aim was to develop a scheme for predicting the refractive index of atmospheric aerosols. They evaluated the proposed scheme using laboratory-generated SOA and even included results from field observations. To our knowledge, this is the only available approach that predicts refractive index based on H/C and O/C ratios. Therefore, we believe it is appropriate to include comparisons with their results, as they help illustrate that the parameterization of organic aerosol refractive index remains an open and unresolved issue.

We also agree with the reviewer that it is still unclear how much the formation pathway

contributes. However, our previous publication has already demonstrated the prevalence of multiphase SOA formation in the Guangzhou urban area, which likely explains this finding.

**References:**

Cai, J., Chu, B., Yao, L., Yan, C., Heikkinen, L. M., Zheng, F., Li, C., Fan, X., Zhang, S., Yang, D., Wang, Y., Kokkonen, T. V., Chan, T., Zhou, Y., Dada, L., Liu, Y., He, H., Paasonen, P., Kujansuu, J. T., Petäjä, T., Mohr, C., Kangasluoma, J., Bianchi, F., Sun, Y., Croteau, P. L., Worsnop, D. R., Kerminen, V. M., Du, W., Kulmala, M., and Daellenbach, K. R.: Size-segregated particle number and mass concentrations from different emission sources in urban Beijing, Atmos. Chem. Phys., 20, 12721-12740, 10.5194/acp-20-12721-2020, 2020.

He, Q., Bluvshtein, N., Segev, L., Meidan, D., Flores, J. M., Brown, S. S., Brune, W., and Rudich, Y.: Evolution of the Complex Refractive Index of Secondary Organic Aerosols during Atmospheric Aging, Environmental science & technology, 52, 3456-3465, 10.1021/acs.est.7b05742, 2018.

He, Q., Li, C., Siemens, K., Morales, A. C., Hettiyadura, A. P. S., Laskin, A., and Rudich, Y.: Optical Properties of Secondary Organic Aerosol Produced by Photooxidation of Naphthalene under NOx Condition, Environmental science & technology, 56, 4816-4827, 10.1021/acs.est.1c07328, 2022.

Li, C., He, Q., Schade, J., Passig, J., Zimmermann, R., Meidan, D., Laskin, A., and Rudich, Y.: Dynamic changes in optical and chemical properties of tar ball aerosols by atmospheric photochemical aging, Atmos. Chem. Phys., 19, 139-163, 10.5194/acp-19-139-2019, 2019.

Li, K., Li, J., Liggio, J., Wang, W., Ge, M., Liu, Q., Guo, Y., Tong, S., Li, J., Peng, C., Jing, B., Wang, D., and Fu, P.: Enhanced Light Scattering of Secondary Organic Aerosols by Multiphase Reactions, Environmental science & technology, 51, 1285-1292, 10.1021/acs.est.6b03229, 2017.

Moise, T., Flores, J. M., and Rudich, Y.: Optical Properties of Secondary Organic Aerosols and Their Changes by Chemical Processes, Chemical Reviews, 115, 4400-4439, 10.1021/cr5005259, 2015.

Zhai, M., Kuang, Y., Liu, L., He, Y., Luo, B., Xu, W., Tao, J., Zou, Y., Li, F., Yin, C., Li, C., Xu, H., and Deng, X.: Insights into characteristics and formation mechanisms of secondary organic aerosols in the Guangzhou urban area, Atmos. Chem. Phys., 23, 5119-5133, 10.5194/acp-23-5119-2023, 2023.

---

## Author Response (AR2)

**Dear Editor:**

We are very grateful for your careful inspection of our manuscript. All raised comments have been explicitly replied point by point and incorporated into the revision. We also thank two reviewers for their helpful comments which have improved the manuscript.

Thank you very much for your attention.

Sincerely Yours

Ye Kuang

**General comment**:

I would like to thank the authors for their revisions to the manuscript in accordance with the reviewers' suggestions. Before I can proceed with accepting the manuscript for publication, I would greatly appreciate it if the authors could kindly address a few points raised by the reviewers that still require attention.

**Response**: Thanks a lot for your time.

**Minor Comments**:

**Comment:** Line 320: The sentence is still not clear. I would suggest a more linear form as follows: This suggests that the chemical processes responsible for the increase in $mr1064,400$ have minimal influence on the chemical properties of aerosol particles near 235 nm, and that variations in $mr1064,400$ and 323 $mr1064,235$ are governed by different chemical and emission processes.

**Response**: Revised accordingly.

**Comment**: I could not find Figure 1 in the revised version.

**Response**: We have ensured that all figures are appropriately presented in the manuscript.

**Comment**: Line 510: You might revise the sentences to make it clearer. I would suggest: "This is likely associated with that MOOA in Guangzhou urban area is mainly formed through multiphase reactions (Zhai et al., 2023) thus it is expected to have higher $mr$, as demonstrated by Li et al. (2017)."

**Response**: Revised accordingly.

**Comment**: Line 512: it is not clear what the authors want to communicate. Do they want to point out that the results from laboratory studies differ from their observations because most of the laboratory studies focus on gas-phase reactions? Please clarify.

**Response**: Yes, that is exactly what we intend to emphasize. We revise sentences here as: "However, as summarized in Moise et al. (2015), most existing laboratory studies that conducted in the context of gas-phase reactions reveal the increase of O/C would generally decrease $m_r$ at the O/C range of LOOA and MOOA of this study (0.6 to 1.27). The finding here is likely associated with that MOOA in Guangzhou urban area is mainly formed through multiphase reactions (Zhai et al., 2023) thus it is expected to have higher $m_r$, as demonstrated by Li et al. (2017)."

**Comment**: Line 412: one of the reviewers asked if the correlation coefficients of 0.25 and -0.24 could be considered significant. The authors correctly underlined that the significance depends on the number of data points. I suggest to quantify the significance of the correlation with an appropriate statistical test that consider correlation coefficient and population size.

**Response**: P-values from t-tests were added. For both correlation analysis, p-values are below 0.01, indicating that the correlations are statistically significant: "Especially, the $m_{r1064,400}$ showed obviously higher correlations with MOOA (R=0.25, p<0.01) than with LOOA (R=-0.24, p<0.01) (Fig.S11)"